# Evaluation of Neurotropic Activity and Molecular Docking Study of New Derivatives of pyrano[4″,3″:4′,5′]pyrido[3′,2′:4,5]thieno[3,2-*d*]pyrimidines on the Basis of pyrano[3,4-*c*]pyridines

**DOI:** 10.3390/molecules27113380

**Published:** 2022-05-24

**Authors:** Shushanik Sh. Dashyan, Eugene V. Babaev, Ervand G. Paronikyan, Armen G. Ayvazyan, Ruzanna G. Paronikyan, Lernik S. Hunanyan

**Affiliations:** 1Scientific Technological Center of Organic and Pharmaceutical Chemistry of National Academy of Sciences of Republic of Armenia, Ave. Azatutyan 26, Yerevan 0014, Armenia; ervand.paronikyan@mail.ru (E.G.P.); armenayv@gmail.com (A.G.A.); paronikyan.ruzanna@mail.ru (R.G.P.); 2Faculty of Chemistry, Moscow State University, 1, GSP-1, 1-3 Leninskiye Gory, 119991 Moscow, Russia; babaev@org.chem.msu.ru; 3Higher School of Economics, National Research University, 7 Vavilova Str., 117312 Moscow, Russia; 4Laboratory of Structural Bioinformatics, Institute of Biomedicine and Pharmacy, Russian-Armenian University, 123 H. Emin Str., Yerevan 0051, Armenia; lernik.hunanyan@rau.am

**Keywords:** pyrano[3,4-*c*]pyridines, thieno[2,3-*b*]pyridine, thieno[3,2-*d*]pyrimidinones, neurotrophic activity, docking analysis

## Abstract

Background: Heterocyclic compounds and their fused analogs, which contain pharmacophore fragments such as pyridine, thiophene and pyrimidine rings, are of great interest due to their broad spectrum of biological activity. Chemical compounds containing two or more pharmacophore groups due to additional interactions with active receptor centers usually enhance biological activity and can even lead to a new type of activity. The search for new effective neurotropic drugs in the series of derivatives of heterocycles containing pharmacophore groups in organic, bioorganic and medical chemistry is a serious problem. Methods: Modern methodology of drugs involves synthesis, physicochemical study, molecular modeling and selection of active compounds through virtual screening and experimental evaluation of the biological activity of new chimeric compounds with pharmacophore fragments. For the synthesis of new compounds, classical organic methods were used and developed. For the evaluation of neurotropic activity of new synthesized compounds, some biological methods were used according to indicators characterizing anticonvulsant, sedative and antianxiety activity as well as side effects. For docking analysis, various soft ware packages and methods were used. Results: As a result of multistep reactions, 11 new, tri- and tetracyclic heterocyclic systems were obtained. The studied compounds exhibit protection against pentylenetetrazole (PTZ) seizures as well as some psychotropic effects. The biological assays evidenced that nine of the eleven studied compounds showed a high anticonvulsant activity by antagonism with pentylenetetrazole. The toxicity of the compounds is low, and they do not induce muscle relaxation in the studied doses. According to the study of psychotropic activity, it was found that the selected compounds have an activating behavior and anxiolytic effects on the “open field” and “elevated plus maze” (EPM) models. The data obtained indicate the anxiolytic (antianxiety) activity of the derivatives of tricyclic thieno[2,3-*b*]pyridines and tetracyclic pyridothieno[3,2-*d*]pyrimidin-8-ones, especially pronounced in compounds **3b**–**f** and **4e**. The studied compounds increase the latent time of first immobilization on the “forced swimming” (FS) model and exhibit antidepressant effects; compounds **3e** and **3f** especially exhibit these effects, similarly to diazepam. Docking studies revealed that compounds **3c** and **4b** bound tightly in the active site of γ-aminobutyric acid type A (GABA_A_) receptors with a value of the scoring function that estimates free energy of binding (∆G) at −10.0 ± 5 kcal/mol. Compound **4e** showed the best affinity ((∆G) at −11.0 ± 0.54 kcal/mol) and seems to be an inhibitor of serotonin (SERT) transporter. Compounds **3c**–**f** and **4e** practically bound with the groove of T4L of 5HT_1A and blocked it completely, while the best affinity observed was in compound **3f** ((∆G) at −9.3 ± 0.46 kcal/mol). Conclusions: The selected compounds have an anticonvulsant, activating behavior and anxiolytic effects and at the same time exhibit antidepressant effects.

## 1. Introduction

In recent years, there has been an increase in the level of natural and social disasters in the world, which has affected people’s health. Patients of the COVID-19 pandemic have respiratory problems, severe pneumonia and neuropsychiatric disorders. At the same time, the number of malignant neoplasms and infectious diseases is increasing year by year. Pharmacological practice now uses drugs, particularly psychotropics, which are not without flaws; these drugs are endowed with toxicity and have an adverse effect on the body. Synthesis of pharmaceuticals for the treatment of neuropsychiatric disorders, in particular epilepsy, is a serious challenge for synthetic organic chemistry. The antiepileptic agents most commonly used in medicine often cause toxic side responses from different organs and systems, emotional disturbances, impaired memory, etc. In this regard, the search for and study of anticonvulsants possessing combined psychotropic properties are of unquestionable interest. In experimental psychopharmacology, while searching for new neurotrophic compounds, it is important and relevant to model both the pathology itself and its individual manifestations in animals. Such an approach of differentiated (application of interoceptive stimuli, such as corazole) and integrative (for example, “open field”) modelling, biostatistical evaluation of the spectrum of pharmacological action of substances and comparison of the major and side effects make it possible to carry out a more detailed selection of promising pharmaceutical agents among newly synthesized compounds. 

Currently, molecular modeling methods and, in particular, molecular docking are intensively used in modern pharmaceutical chemistry for the study and primary assessment of the bioactivity of newly synthesized and/or modified compounds [1]. Moreover, one of the promising strategies in creating new pharmaceuticals is the design and synthesis of hybrid compounds consisting of two or more different bioactive fragments acting through the activation of several mechanisms of action [2].Docking analysis is basically used for the conformational search for the best and most reliable orientation of ligand during complexation of the ligand target [3]. To achieve the maximal result (approximate to real conditions), pair docking is usually used for predicting the interaction of the ligand with the target. 

The derivatives of condensed pyridines are of interest as biologically active substances [4]. Thus, a large number of substituted pyrazolopyridine derivatives have been found to possess various biological properties such as A(1) adenosine receptor antagonism [5], antimicrobial properties [6], nonanionic antiplatelet agents [7] and psychotropic effects [8]. Some alkaloids of the pyrano[3,4-*c*]pyridine series exert universal effects: hypotensive, anticonvulsant, antipsychotic, anti-inflammatory and antitumor effects [9,10,11]. The derivatives of fused pyrimidines have also attracted a considerable interest in medicinal chemistry research due to their versatility and a broad bioactive potential [12]. Thieno[3,2-*d*]pyrimidines are structural analogues of purines. Purines, as endogenous scaffolds, play an important biochemical role in a variety of regular physiological functions. As bioisosteres to purines, thieno[3,2-*d*]pyrimidines were also found to exhibit numerous biological activities, probably due to their interaction with various physiological factors [13,14]. On the other hand, tetracyclic fused systems containing pyran, pyridine, thiophene and pyrimidine rings can be considered analogues of heterosteroids, which are known to display varied biological effects [15,16,17]. Moreover, literature data evidenced that bi-, tri-, tetra- and pentacyclic systems containing pyridine and pyrimidine rings were endowed with neurotropic properties [18,19,20], and the addition of a cycle does not reduce biological activity.

In a previous paper [21], we described the synthesis as well as the neurotropic activity of a series of derivatives of pyrazol-1-yl-substituted pyrano[3,4-*c*]pyridine. The results of these previous studies have enabled us to identify several compounds that exhibited potent and wide-spectrum anticonvulsant properties in maximal electroshock seizure (MES) and subcutaneous pentylentetrazole (PTZ) seizure tests. Furthermore, the compounds were studied for their anxiolytic and antidepressant activities in some psychotropic models, such as “open field”, ″elevated plus-maze″ (EPM) and ″forced swimming″ (FS). Moreover, our studies have shown that three derivatives among the pyrazol-1-yl-substituted pyrano[3,4-*c*]pyridines **1** and **2** exhibited anticonvulsant, anxiolytic, some antidepressant and sedative activities similar to the commercial drug diazepam; this also indicates activity (Figure 1).

Based on above considerations we now synthesized new compounds with the aim of evaluating neurotropic activity and probable mechanisms of action and to study the influence of the structure of compounds on biological activity. Continuing our studies in the field, in this paper, we present the synthesis of new pyrazol-1-yl-substituted tricyclic and tetracyclic heterocyclic systems—*N*-alkyl(aryl) derivatives of pyrano[4,3-*d*]thieno[2,3-*b*]pyridine and pyrano[4″,3″:4′,5′]pyrido[3′,2′:4,5]thieno[3,2-*d*] pyrimidin-8(9*H*)-one. We also study the neurotropic properties of the synthesized compounds and perform molecular docking to evaluate probable mechanisms of action.

## 2. Results and Discussion

### 2.1. Chemistry

For the synthesis of the target compounds, we used as starting materials 8-(3,5-dimethyl-1*H*-pyrazol-1-yl)-3,3-dimethyl-6-thioxo-3,4,6,7-tetrahydro-1*H*-pyrano[3,4-*c*]pyridine-5-carbonitrile (**1**) and ethyl {[5-cyano-8-(3,5-dimethyl-1*H*-pyrazol-1-yl)-3,3-di methyl-3,4-dihydro-1*H*-pyrano[3,4-*c*]pyridin-6-yl]thio}acetate (**2**) (Figure 1).

Both starting materials **1** and **2** are versatile substrates: in fact, both of them are decorated on nearby carbon atoms of the pyridine ring by two functional groups (nitrile and thioxo-/thioalkyl) able to open the way to compounds with a new fused ring (a thiophene ring), which, in turn, will still contain other reactive groups useful for further chemical transformation.

Additionally, 8-(3,5-Dimethyl-1*H*-pyrazol-1-yl)-3,3-dimethyl-6-thioxo-3,4,6,7-tetra hydro-1*H*-pyrano[3,4-*c*]pyridine-5-carbonitrile (**1**) and ethyl {[5-cyano-8-(3,5-dimethyl-1*H*-pyrazol-1-yl)-3,3-dimethyl-3,4-dihydro-1*H*-pyrano[3,4-*c*]pyridin-6-yl]thio}acetate(**2**)were used as key intermediates for the synthesis of a new tricyclic condensed system—*N*-alkyl(aryl) derivatives of pyrano[4,3-*d*]thieno[2,3-*b*]pyridines **3a-f** (yield 78−98%). The physicochemical characterizations of compounds **1** and **2** [21] were already reported.

Compound **2** was subjected to intramolecular cyclization by the action of sodium ethoxide to obtainsubstituted pyrano[4,3-*d*]thieno[2,3-*b*]pyridine **3a** (*method A*). The latest was also synthesized in *one pot* from pyridinethione**1** and ethyl chloroacetate in the presence of sodium ethoxide (*method B*). In the case of alkylation of compound **1** with the chloroacetic acid derivatives which contain an active methylene group, the closure of the thiophenic ring proceeds simultaneously with the alkylation. As a result of *one-pot* reaction, new tricyclic condensed system compounds **3a**–**f** were synthesized with good yields (Figure 1, Table 1).

The structures of newly synthesized compounds **3a**–**f** were confirmed by NMR, IR spectroscopy and by elemental analysis. Thus, in the ^1^H-NMR spectra of these new compounds, the presence of the NH_2_ group proton at 6.25−7.00 ppm was observed (see Appendix A). The IR spectra of **3a**–**f** show the amino group (NH, NH_2_) absorptions near 3238−3504 cm^−1^.

The newly synthesized 1-amino-*N*-alkyl(aryl)-5-pyrazol-1-yl-pyrano[4,3-*d*] thieno[2,3-*b*]pyridines **3a,c**–**f** still contain amino and carboxy functional groups which can undergo new cyclization reactions. Thus, by interaction of compound **3a** with formamide pyrano[4″,3″:4′,5′]pyrido[3′,2′:4,5]thieno[3,2-*d*]pyrimidin-8-one, **4a** was synthesized. By condensation of compounds **3c**–**f** with triethylorthoformate in the presence of acetic anhydride, new derivatives of pyrano[4″,3″:4′,5′]pyrido[3′,2′:4,5]thieno[3,2-*d*] pyrimidin-8-ones **4b**–**e** were synthesized with high yields (Figure 2, Table 2).

The structures of newly synthesized pyrano[4″,3″:4′,5′]pyrido[3′,2′:4,5] thieno[3,2-*d*] pyrimidin-8-ones **4a**–**e** were supported by NMR and IR spectroscopy. Thus, in the ^1^H-NMR spectra, the signals of the NH_2_ group characteristic of the initial compounds **3** were absent, while the pyrimidine ring CH group signals appeared, indicating the cyclization of compounds **4**. The structure of compounds **4a**–**e** was also supported by ^13^C-NMR data. In ^13^C-NMR spectra, the signals of CH and C=O groups of the pyrimidine cycle are observed at 146.4–149.0 and 157.5–161.2 ppm, respectively (see Appendix A). The IR spectra of compounds **4a**–**e** did not show the characteristic bands of the amino group but showed bands in the range of ν 1668–1675 cm^−1^, typical for the carbonyl group.

### 2.2. Biological Assays

The neurotropic activity of 11 newly synthesized heterocyclic compounds—5-pyrazol-1-yl substituted pyrano[4,3-*d*]thieno[2,3-*b*]pyridines **3a**–**f** and pyrano[4″,3″:4′,5′]pyrido[3′,2′:4,5]thieno[3,2-*d*]pyrimidin-8-ones **4a**–**e**—was carried out according to indicators characterizing anticonvulsant, sedative and antianxiety activity and side effects.

The anticonvulsive action of the tested compounds was assessed by evaluating the antagonism between the convulsive pentylenetetrazole (PTZ) action and maximal electroshock seizures (MES) [22,23,24,25,26]. PTZ-induced testing is considered an experimental model for the clonic component of epilepsy seizures and prognostic anxiolytic [27] activities of the compounds. The MES test is used as an animal model for the generalized tonic seizures of epilepsy. Ethosuximide was used as a control [28]. The side effects of the compounds—neurotoxicity (movement coordination disorder, myorelaxation and ataxia) with the test of ″rotating rod″ [22,29] and maximal tolerated dose (MTD)—were also studied on mice. To determine the 50% effective and 50% neurotoxic doses, a statistical method of probit analysis by Litchfield and Wilcoxon was used [30,31]. From a practical point of view, the active compound’s protective index (PI) was identified (Protective Index = Toxic Dose of 50% (TD_50_)/Effective Dose of 50% (ED_50_)).

Evaluation of the anticonvulsant activity of all the synthesized compounds revealed that they, to varying degrees, exhibit PTZ antagonism. Thus, the compounds, at a dose of 50 mg/kg, prevented PTZ clonic seizures in 40–80% of animals. However, compounds **3b**–**f** and **4b**–**e** had a pronounced anticonvulsant action. Intraperitoneal injections of these compounds into mice, starting with a dose of 20 mg/kg, were accompanied by the prevention of PTZ seizures, and the ED**_50_** ranged from 24 mg/kg to 44 mg/kg (Table 3). It should be mentioned that tested compounds are more active than ethosuximideaccording to the test on PTZ, but less so than diazepam. The effective dose of ethosuximide (ED**_50_**, mg/kg) in the antagonism with PTZ in mice was 155.0 mg/kg, while for diazepam it was 0.5 mg/kg. The compounds, tested by the ″rotating rod″ method, in doses of 50–100 mg/kg in mice did not violate the coordination of movements; no signs of muscle relaxation were observed. TD**_50_** of the studied compounds ranged from 505 mg/kg to 600 mg/kg. Ethosuximide in the studied doses of 100–200 mg/kg in mice also does not cause muscle relaxation.

The structure–activity relationship study revealed that the presence of 3,4-dichlorophenyl substituent on the second position of thiophene ring of tricyclic pyranothienopiridine moiety (**3e**) is beneficial for anticonvulsant activity. The replacement of 3,4-dichlorophenyl substituent by carbonitrile (**3b**), phenyl (**3c**), 2,4-dimethoxyphenyl (**3d**) or phenylethyl (**3f**) groups slightly decreased the activity, while replacement of the 3,4-dichlorophenyl group with an ethyl carboxylate group (**3a**) decreased the activity much more. Similar activity is observed in the case of tetracyclic thienopyrimidines.

According to the maximal tolerated doses (MES) test, the compounds studied, as well as reference drugs, did not exhibit an anticonvulsant effect. They did not protect from tonic and clonic seizures caused by MES. Maximal tolerated doses of studied compounds and ethosuximide are within the limits of 1000−1500 mg/kg. Protective indexes of the selected compounds were large, especially for compound **3e**, and far exceed indexes of ethosuximide and diazepam.

The most effective 9 compounds—**3b**–**f** and **4b**–**e**—were studied on the ″open field″, ″elevated plus maze″ (EPM) and ″forced swimming″ tests at a dose of 50 mg/kg since the ED**_50_**s of these compounds are within 50 mg/kg at the confidence intervals.

In the ″open field″ behavioral model, [32,33,34] in rats of the control group, the number of horizontal displacements, vertical displacements and examined cells were 25.8, 6.1 and 0.5, respectively (Table 4).

The compounds under study cause changes in the behavioral indices in comparison with the control—with the injection of the compounds, marked changes in the horizontal and vertical movements of animals were observed. Compounds **3c**–**f** and **4b**–**e** statistically significantly increase the horizontal movements of animals; others do not result in any behavior changes associated with horizontal movements. As for vertical movements, compounds **3b** and **3e** statistically significantly reduce them, exhibiting some sedative effect. In contrast, compound **3d** increases vertical movements and thereby leads to activation of behavior. However, all selected compounds statistically significantly, compared with the control, increase the number of sniffing cell examinations, which may be due to the manifestation of the antianxiety activity of the compounds (Table 4), which was especially pronounced in compounds **3b, 3d, 3e, 3f** and **4e**.Ethosuximide at the effective dose of 200 mg/kg has no effect on all indicators of research activities, while diazepam (2 mg/kg), in comparison with the control group of mice, causes a significant increase in the number of cells examined, i.e., pronounced antianxiety effect. The studied compounds **3c, 3d, 3f, 4b**–**e** and diazepam, by increasing horizontal movements, exhibit an activating effect on behavior.

In order to assess fears, the methodology of ″elevated plus maze″ (EPM) developed by Pellow was used [35]. The ″elevated plus maze″ is a behavioral assay (fear) used to estimate the antianxiety effects of pharmacological agents, synthetic compounds, etc. [36,37,38]. In brief, rats or mice are placed at the junction of the four arms of a maze with their face to an open arm, followed by the recording of their entries/duration in each arm by a video tracking system and observer simultaneously for 5 min. On the EPM model, control animals are predominantly in closed sleeves (Table 5).

After administration, compounds **3b**–**f** and **4b**–**e** statistically reliably decrease the time spent in closed arms, and after intraperitoneal injection of compounds **3d**, **3f** and **4d**, **a** decrease in the number of entries into the closed arms was observed. All selected compounds were statistically significant and, compared to the control, increased the time spent by experienced animals in the center, which indicates sedative activity, especially in compounds **3e**, **3f** and **4c**. After injection of the compounds, experimental animals went to the open arms and were located there for 3 (**4c**), 9.8 (**3e**), 10 (**4d**) and 11.4 (**3d**) *s* in contrast to the control.

The time spent in the open arms by mice after administration of compound **3d** was 11.4 s; at the same time, control mice which received ethosuximide at a dose of 200 mg/kg did not enter the open arms due to fear. Animals that received diazepam at a dose of 2 mg/kg also entered the open arms and stayed there for 57 s. The data obtained indicate anxiolytic activity in all of the selected compounds, especially expressed in compounds **3e, 3d, 3f** and **4d**.

The structure–activity relationships study revealed that the presence of phenyl (**3c**), 2,4-dimethoxyphenyl (**3d**) and phenethyl (**3f**) substituent on the second position of the thiophene ring of tricyclic pyranothienopiridine moiety and all tetracyclic pyrano[4″,3″:4′,5′]pyrido[3′,2′:4,5]thieno[3,2-*d*]pyrimidin-8-ones (**4b-e**) except compound **4a** increased the horizontal movements of animals, exhibiting an activating effect. Tricyclic thienopyridines containing nitrile (**3b**) or 3,4-dichlorophenyl (**3e**) substituents on the second position of the thiophene ring decreased the vertical movements of animals, exhibiting some sedative effect. The compound (**3d**) 2,4-dimethoxyphenyl substituent is beneficial for its activating effects of both horizontal and vertical movements.

The ″forced swimming″ test (FS) is one of the most commonly used assays [39]. The FST is used to monitor depressive-like behavior and is based on the assumption that immobility reflects a measure of behavioral despair.

On the ″forced swimming″ model in control mice, the first immobilization occurs after 92 s (Table 6). Some selected compounds (**3e, 3f**) tested at a dose of 50 mg/kg statistically significantly increased the duration of the latent period of the first immobilization and decreased the total immobilization duration.

For compounds **3e** and **3f,** the duration of the latent period of the first immobilization increased to 172 and 121 s, respectively, while the total time of immobilization decreased to 48 and 20 s, respectively. This suggests that compounds **3e** and **3f** studied at a dose of 50 mg/kg show some antidepressant effect in the same manner as diazepam. These compounds and diazepam increase the total time of active swimming, but not statistically significantly. The remaining compounds decrease the total time of active swimming. The data obtained with the use of ethosuximide at a dose of 200 mg/kg coincide with the control data.

According to the structure–activity relationship study, the presence of 3,4-dichlorophenyl (**3e**) or phenylethyl (**3f**) substituent on the second position of the thiophene ring of tricyclic thienopyridine moiety as well as 3,4-dichlorophenyl (**4d**) substituent on the ninth position of the pyrimidine ring of tetracyclic pyrano[4″,3″:4′,5′]pyrido[3′,2′:4,5]thieno[3,2-*d*]pyrimidinone moiety is favorable for antidepressant activity.

### 2.3. Molecular Docking

Typically, antiepileptic drugs block sodium channels or enhance γ-aminobutyric acid type A (GABA) function by general targeting of GABA_A_ receptors and, therefore, new compounds docked with the GABA_A_ receptor. The serotonin transporter (SERT or 5HT_1A) is implicated in a number of neurobehavioral disorders (e.g., depression, anxiety, autism). The results of in vivo investigation showed that some of the newly synthesized compounds were endowed with antidepression and antianxiety properties and therefore docked with the SERT or 5HT_1A receptors.

#### 2.3.1. Docking and Conformational Analysis of GABA_A_ Receptor Complexation

Docking analysis of the studied compounds revealed that of 11 compounds, the interaction of seven (**3a**, **3c**, **3d**, **3e**, **4b**, **4c** and **4d**) with the GABA_A_ receptor was observed. To compare the effects of the test compounds with GABA_A_, diazepam was used as a control. The results of docking analysis indicate that diazepam interacts by hydrophobic and electrostatic forces with GABA_A_ in subsite 1 of the ECD (extracellular domain) interface with an energy of −7.5 kcal/mol (Table 7).

Compounds **3b, 3f, 4a** and **4e** do not bind to this receptor. Complexation is observed both in the active center of the protein and in allosteric sites, which are evidenced by the obtained spatial interaction parameters. Energy values of complexation were calculated for seven compounds. The lowest binding energy was found for compounds **3c** and **4b**, followed by **3d,** but compound **3a** had the highest binding energy (Table 7). The mean square deviation during the complexation did not exceed RMSD ≤ 2 Å. Obtaining the spatial parameters of complexation is evidence that all the investigated compounds bind to the ligand binding domain in two basic sites of the protein. A conformational map of the binding of diazepam to GABA_A_ indicates that the amino acid residues Phe200, Tyr205, Ala201, Tyr 97, Glu155 and Tyr202 in chain C and Asp 43, Tyr176, Tyr66, Asn41, Gln64 and Tyr 62 in chain B are involved in the complexation process (Figure 2).

The first is the benzamidine site of subsite 1 of the ECD interface [40]. This loop is formed due to amino acid residues included in chains B and C, as well as in E and D [41]. Compounds **3a, 3c** and **3e** are binding at this site. A study of the interaction of **3a** with GABA_A_ revealed that complexation is due to hydrophobic interactions and hydrogen bonds (Figure 3). Hydrogen bonds are observed with the amino acid residues in the chain C by Tyr 202 and Tyr97 and in the B chain by Tyr 62. The length of the observed hydrogen bonds does not exceed 3.0 Å.

In the case of **3c**, the hydrogen bonds were not detected at the interaction. The interaction is carried out at the expense of electrostatic and hydrophobic interactions (Figure 4).

Compound **3e** binds in chain B and C through both hydrogen and electrostatic interactions; hydrophobic interactions are also present. From the point of view of spatial arrangement, it practically repeats the position of **3a**, while hydrogen bonds are formed with Tyr97 in chain C and Tyr 62 in chain B, not exceeding a value of 3.3 Å. The values of the binding constants can be explained by the difference in electrostatic forces and the amount of amino acid residue in favor of **3e** (Figure 5).

The second site of the investigated compounds is subsite 3 in the ECD interface, which is specific for cations and responsible for inhibition. This site is usually formed between the α+/β subunit of chains with the involvement of amino acid residues in the positions of 137, 127 and 182 [42]. Complexation at this site is observed by four compounds (**3d, 4b, 4c** and **4d**). The interaction of **3d** is due to the formation of a hydrogen bond with Met 137 with a distance of 3.3 Å as well as electrostatic interaction. In other cases, there is a hydrophobic interaction (Figure 6).

The spatial orientation of **4b** is different from position **3d,** with a deviation of 5.8 Å, herewith there is observed a hydrogen bond with Val 50, linked by the binding site of the subsite 3. Compound **4b** binds with Met137 from the key residues and exhibits electrostatic and hydrophobic interaction with Glu182. Despite the absence of hydrogen bonds with the key residues of subsite 3, the advantage of **4b** is obvious compared to **3d** in terms of spatial-energy characteristics.

Obtained spatial characteristics of **4c** and **4d** have shown that they practically repeated the position of **4b** with a deviation of 0.8 Å and the same types of interaction. The difference is the absence of hydrogen bonds with the residue in subsite 3. The interaction is carried out by the electrostatic and hydrophobic interactions with identical amino acid residues. From this point of view, it can be predicted that compounds **4b**, **4c** and **4d** will be identical; at the same time, the highest-binding constant values are exhibited by compounds **4b** and **4c**.

#### 2.3.2. Docking and Conformational Analysis of SERT Complexation

A study of the interaction of the compounds with SERT (serotonin transporter) revealed that the eight compounds interact with SERT, with the exception of **2**, **3a, 4a** and **4c**. The biophysical properties of interaction for eight compounds have been calculated and are presented in Table 8.

Obtained spatial and conformation parameters of complexation are evidence that the studied compounds’ positions can be divided into two classes. The first class includes compounds for which binding occurs especially at the central site of the active center of SERT [43], such as **2**, **3e** and **4e**. The second class is formed by compounds that bind predominantly in the allosteric site [44], such as **3c**, **3d**, **3f**, **4b** and **4d**. Compounds that bind at the central site of SERT exhibit different energy indices and binding constants. Compound **4e** possesses the highest values, with the involvement of key amino acid residues Ile172, Tyr176, Phe335, Tyr95, Phe341 and Val501 [45]. The spatial arrangement of **2** corresponds to the position of **4e** with a deviation at a38° angle. At the same time, the amino acid residues involved in the process of complexation correspond to each other with a difference of just one residue. Compound **2,** instead of Val501, has Tyr497, which is also included in the central site [46]. The conformational maps of complexation obtained indicate that interactions carried out by hydrogen bonds and electrostatic forces. Hydrophobic interactions are also involved in the complexation. In compound **2,** the hydrogen bonds are formed with two residues, the Tyr497 and Glu 493, which do not reach the central binding site with distances of 3.3 and 3.0 Å, respectively. In **4e** one hydrogen bond is visualized with Tyr175, which is not included in the active site. It should be noted that the interaction of **4e** in the central site of the active center is predominantly carried out due to electrostatic and hydrophobic forces. The basically spatial position of **3e** is different from the rest of other members of the group. The quantity of amino acid residues involved in the complexation is less than that of **2** and **4e**. Compound **3e** interacts with Ile172, Tyr176, Phe335 and Tyr497 and formed a hydrogen bond with Tyr497 with a distance of 3.0 Å. In other cases, hydrophobic interaction predominates. It should be noted that **3e** also interacts hydrophobically with the amino acid residues Phe556 and Pro561 of the allosteric site [47]. Single hydrophobic interactions with Arg104 are also observed in **2** and **4e**. The results of superposition of **2**, **3e** and **4e** in the central site are presented in Figure 7.

The spatial positions and binding types of compounds **3c**, **3d**, **3f**, **4b** and **4d** in the allosteric site can be divided into three sub-classes. The first sub-class includes **3d** and **4d;** the second sub-class includes **3c** and **3f;** the third sub-class includes only **4b**. The last is unique because it exhibits a directed action without affecting the central binding site of SERT (Figure 8).

In this case, predominantly hydrophobic interactions with practically all residues are observed, forming an allosteric binding site. Such residues include Arg104, Asp328, Ala331, Phe556, Pro561 and Glu494. The second sub-class compounds (**3c, 3f**), by the values of conformation, are identical and interactions are carried out due to hydrophobic and electrostatic interactions involving amino acid residues Arg104, Ala33, Glu494 and Phe556. In addition to the allosteric site, hydrophobic interactions with Phe335 are observed (Figure 9).

The positions in the allosteric site of the best conformers of the first subclass are located mirrored relative to each other. The difference of root-mean-square deviation (RMSD) is 0.5 Å with an angle of 36°. Amino acid residues Arg104, Asp328, Ala331 and Phe556 are included in the complexation for both representatives of this subclass. The interactions are carried out at the expense of hydrophobic interaction. Apart from that, hydrophobic interaction of Phe335 and Tyr176 with the central binding site is observed in **4d**, and in **3d,** it is observed forTyr176 and Ile172 (Figure 10).

#### 2.3.3. Docking and Conformational Analysis of 5HT_1A Complexation

The results of docking analysis for investigated compounds with 5HT_1A exhibited that interactions are observed for seven compounds (Table 9).

For compounds **3b, 4b, 4c** and **4d** no interaction was observed. All compounds except **3a** and **4a** show similar results of biophysical indicators of interaction. The obtained conformational maps indicate that the interaction is carried out in two clusters with the involvement of the ICL (intracellular loop) of TM5 and TM6 helices [48], where interaction is observed for **3a** and **4a** (Figure 11).

The second is a cluster in the T4L (T4-lysozyme) domain [49], where compounds **3c**, **3d**, **3e**, **3f** and **4e** bind. The spatial positions of **3a** and **4a** in the ICL are practically identical to RMSD 0.5 Å with an angle of 7.2°. The interaction of **3a** and **4a** is due to the van der Waals forces and hydrophobic interactions with the key residues of Arg131, Phe332, Glu268and Tyr66, although **3a** has two hydrogen bonds with Ser143 and Val 67 (Figure 11).

For compounds included in the second cluster, the interactions in the T4L domain are carried out basically due to hydrophobic force with the involvement of amino acid residues Glu1011 and Leu1032. It should be noted that compound **3f** forms ahydrogen bond with Glu1011 with a distance of 3.2 Å. All compounds also interact with Asp1020 due to hydrophobic forces, except for**3e**. In the same way, interactions are observed with Tyr1018, except for **3c**. Compounds **3e** and **3f** form hydrophobic bonds with Arg1014 and Glu1022 (Figure 12).

The second cluster compounds practically bind with the groove of T4L and block it completely [50]. At the same time, they show similar results of energy parameters. The best compound, judging by the indicators, is **3f**.

## 3. Materials and Methods

### 3.1. Chemistry

#### 3.1.1. General Information

All chemicals and solvents were of commercially high purity grade purchased from Sigma-Aldrich (Saint Louis, MO, USA). Melting points (m.p.) were determined on a Boetius microtable. They are expressed in degrees centigrade (°C). ^1^H NMR and ^13^C NMR spectra were recorded in DMSO-*d_6_*/CCl_4_, 1/3, *v*/*v* solution (300MHz for ^1^H and 75.462 MHz for ^13^C) on a Varian mercury spectrometer (Varian Inc., Palo Alto, CA, USA). Chemical shifts are reported as δ (parts per million) relative to TMS (tetramethylsilane) as the internal standard. IR spectra were recorded on Nicolet Avatar 330-FTIR spectrophotometer (Thermo Nicolet, Foster, CA, USA) and the reported wave numbers are given in cm^−1^. Elemental analyses were performed on a Euro EA 3000 Elemental Analyzer (EuroVector, Pavia, Italy).

#### 3.1.2. Methods for the Synthesis of Ethyl 1-Amino-5-(3,5-dimethyl-1*H*-pyrazol-1-yl)-8,8-dimethyl-8,9-dihydro-6*H*-pyrano[4,3-*d*]thieno[2,3-*b*]pyridine-2-carboxylate (**3a**)

*Method A:* To a solution of sodium ethoxide prepared from sodium (0.046 g, 2 mmol) and ethanol (20 mL), compound 2 (0.8 g, 2.0 mmol) was added. The mixture was stirred for 2 h at 60 °C and cooled; the white precipitate was filtered off and washed with water, dried and recrystallized from EtOH. Yield 81%, m.p. 160–161 °C.IR ν/cm^−1^: 3504, 3370 (NH_2_), 1664 (C=O). ^1^H NMR (300 MHz, DMSO-*d_6_*/CCl_4_, 1/3, *v*/*v*) δ_H_: 1.29 (t, *J* = 7.1 Hz, 3H, OCH_2_CH_3_), 1.30 (s, 6H, C(CH_3_)_2_), 2.19 (s, 3H, CH_3_), 2.27 (s, 3H, CH_3_), 3.32 (s, 2H, 9-CH_2_), 4.29 (q, *J* = 7.1 Hz, 2H, OCH_2_), 4.67 (s, 2H, 6-CH_2_), 6.11 (s, 1H, CH), 6.87 (s, 2H, NH_2_). ^13^C NMR (75.462 MHz, DMSO-*d_6_*/CCl_4_, 1/3, *v*/*v*) δ_C_: 11.8, 13.3, 14.3, 26.3, 36.4, 59.4, 60.2, 69.1, 107.2, 121.1, 123.1, 141.1, 144.3, 148.2, 148.7, 149.5, 156.4, 164.5. Anal. calcd for C_20_H_24_N_4_O_3_S: C 59.98; H 6.04; N 13.99; S 8.01%. Found: C 60.13; H 5.98; N 14.13; S 7.85%.

*Method B:* To a solution of sodium ethoxide prepared from sodium (0.09 g, 4.0 mmol) and anhydrous ethanol (30 mL), compound 1 (0.63 g, 2.0 mmol) was added. The mixture was stirred until it became homogeneous, then ethyl chloroacetate (0.25 g, 2 mmol) was added, the mixture was stirred for 2 h at 60 °C and cooled and the white precipitate was filtered off, washed with water and recrystallized from EtOH. Yield 78%.

#### 3.1.3. General Method for the Preparation of Compounds **3b**–**f**

To a solution of anhydrous potassium carbonate (0.69 g, 5.0 mmol) in ethanol (20 mL), compound 1 (0.63 g, 2.0 mmol) and the appropriate alkylating agent (2.0 mmol) were added. The mixture was refluxed for 3 h. After cooling, the obtained crystals were filtered off, washed with water, dried and recrystallized from ethanol/dioxane mixture (1:1).

*1-Amino-5-(3,5-dimethyl-1H-pyrazol-1-yl)-8,8-dimethyl-8,9-dihydro-6H-pyrano[4,3-d]thieno[2,3-b]pyridine-2-carbonitrile* (**3b**). It was obtained as white crystals; yield 89%, m.p. 260–261 °C. IR ν/cm^−1^: 3478, 3332, 3238 (NH_2_), 2198 (CN). ^1^H NMR (300 MHz, DMSO-*d_6_*/CCl_4_, 1/3, *v*/*v*) δ_H_:1.34 (s, 6H, C(CH_3_)_2_), 2.23 (s, 3H, CH_3_), 2.35 (s, 3H, CH_3_), 3.29 (t, *J* = 1.3 Hz, 2H, 9-CH_2_), 4.71 (t, *J* = 1.3 Hz, 2H, 6-CH_2_), 5.96 (s, 1H, CH), 6.25 (br s, 2H, NH_2_). ^13^C NMR (75.462 MHz, DMSO-*d_6_*/CCl_4_, 1/3, *v*/*v*) δ_C_: 11.9, 13.1, 26.2, 36.2, 59.5, 68.7, 75.5, 106.9, 114.7, 121.2, 121.5, 140.6, 143.6, 147.9, 148.2, 151.0, 156.4. Anal. calcd for C_18_H_19_N_5_OS: C 61.17; H 5.42; N 19.81; S 9.07%. Found: C 61.32; H 5.34; N 19.67; S 9.25%.

*1-Amino-5-(3,5-dimethyl-1H-pyrazol-1-yl)-8,8-dimethyl-N-phenyl-8,9-dihydro-6H-pyrano[4,3-d]thieno[2,3-b]pyridine-2-carboxamide* (**3c**). It was obtained as white crystals; yield 98%, m.p. 220–221 °C.IR ν/cm^−1^: 3453–3281 (NH_2_, NH), 1643 (C=O).^1^H NMR (300 MHz, DMSO-*d_6_*/CCl_4_, 1/3, *v*/*v*) δ_H_: 1.36 (s, 6H, C(CH_3_)_2_), 2.25 (s, 3H, CH_3_), 2.35 (s, 3H, CH_3_), 3.35 (t, *J* = 1.2 Hz, 2H, 9-CH_2_), 4.69 (t, *J* = 1.2 Hz, 2H, 6-CH_2_), 5.96 (s, 1H, CH), 6.90 (br s, 2H, NH_2_), 6.98-7.05 (m, 1H, CH), 7.22-7.30 (m, 2H, 2CH), 7.69-7.74 (m, 2H, 2CH), 9.14 (s, 1H, NH). ^13^C NMR (75.462 MHz, DMSO-*d_6_*/CCl_4_, 1/3, *v*/*v*) δ_C_: 11.8, 13.1, 26.3, 36.4, 59.5, 68.7, 99.3, 106.5, 120.7, 120.9, 122.7, 123.7, 127.6, 138.7, 140.4, 143.2, 147.4, 148.0, 148.3, 155.1, 163.5. Anal. calcd for C_24_H_25_N_5_O_2_S: C 64.41; H 5.63; N 15.65; S 7.16%. Found: C 64.57; H 5.70; N 15.79; S 7.34%.

*1-Amino-N-(2,4-dimethoxyphenyl)-5-(3,5-dimethyl-1H-pyrazol-1-yl)-8,8-dimethyl-8,9-dihydro-6H-pyrano[4,3-d]thieno[2,3-b]pyridine-2-carboxamide* (**3d**). It was obtained as white crystals; yield 92%, m.p. 204–205 °C.IR ν/cm^−1^: 3455–3293 (NH_2_, NH), 1640 (C=O). ^1^H NMR (300 MHz, DMSO-*d_6_*/CCl_4_, 1/3, *v*/*v*) δ_H_: 1.36 (s, 6H, C(CH_3_)_2_), 2.25 (s, 3H, CH_3_), 2.37 (s, 3H, CH_3_), 3.34 (t, *J* = 1.2 Hz, 2H, 9-CH_2_), 3.80 (s, 3H, OCH_3_), 3.93 (s, 3H, OCH_3_), 4.72 (t, *J* = 1.2 Hz, 2H, 6-CH_2_), 5.97 (s, 1H, CH), 6.45 (dd, *J* = 8.8, 2.6 Hz, 1H, CH), 6.54 (d, *J* = 2.6 Hz, 1H, CH), 6.84 (s, 2H, NH_2_), 7.87 (s, 1H, NH), 8.05 (d, *J* = 8.8 Hz, 1H, CH).^13^C NMR (75.462 MHz, DMSO-*d_6_*/CCl_4_, 1/3, *v*/*v*) δ_C_: 11.9, 13.1, 26.3, 36.3, 54.7, 55.4, 59.5, 68.7, 98.1, 99.0, 103.5, 106.7, 120.6, 121.0, 121.1, 123.9, 140.5, 143.4, 147.4, 147.9, 148.0, 149.5, 154.3, 156.0, 162.4. Anal. calcd for C_26_H_29_N_5_O_4_S: C 61.52; H 5.76; N 13.80; S 6.32%. Found: C 61.62; H 5.81; N 13.66; S 6.45%.

*1-Amino-N-(3,4-dichlorophenyl)-5-(3,5-dimethyl-1H-pyrazol-1-yl)-8,8-dimethyl-8,9-dihydro-6H-pyrano[4,3-d]thieno[2,3-b]pyridine-2-carboxamide* (**3e**). It was obtained as white crystals; yield 90%, m.p. 226–227 °C. IR ν/cm^−1^: 3450–3287 (NH_2_, NH), 1643 (C=O). ^1^H NMR (300 MHz, DMSO-*d_6_*/CCl_4_, 1/3, *v*/*v*) δ_H_: 1.36 (s, 6H, C(CH_3_)_2_), 2.24 (s, 3H, CH_3_), 2.35 (s, 3H, CH_3_), 3.34 (t, *J* = 1.2 Hz, 2H, 9-CH_2_), 4.69 (t, *J* = 1.2 Hz, 2H, 6-CH_2_), 5.96 (s, 1H, CH), 7.00 (br s, 2H, NH_2_), 7.36 (d, *J* = 8.8 Hz, 1H, CH), 7.70 (dd, *J* = 8.8, 2.5 Hz, 1H, CH), 8.12 (d, *J* = 2.5 Hz, 1H, CH),9.43 (s, 1H, NH). ^13^C NMR (75.462 MHz, DMSO-*d_6_*/CCl_4_, 1/3, *v*/*v*) δ_C_: 11.8, 13.1, 26.3, 36.4, 59.5, 68.7, 98.2, 106.6, 120.0, 121.0, 121.9, 123.4, 124.9, 129.2, 130.8, 138.9, 140.4, 143.5, 147.6, 148.0, 149.1, 155.3, 163.7. Anal. calcd for C_24_H_23_Cl_2_N_5_O_2_S: C 55.82; H 4.49; N 13.56; S 6.21%. Found: C 55.98; H 4.41; N 13.74; S 6.07%.

*1-Amino-5-(3,5-dimethyl-1H-pyrazol-1-yl)-8,8-dimethyl-N-(2-phenylethyl)-8,9-dihydro-6H-pyrano[4,3-d]thieno[2,3-b]pyridine-2-carboxamide* (**3f**). It was obtained as white crystals; yield 89%, m.p. 97–98 °C. IR ν/cm^−1^: 3409–3285 (NH_2_, NH), 1628 (C=O). ^1^H NMR (300 MHz, DMSO-*d_6_*/CCl_4_, 1/3, *v*/*v*) δ_H_:1.35 (s, 6H, C(CH_3_)_2_), 2.24 (s, 3H, CH_3_), 2.32 (s, 3H, CH_3_), 2.83-2.91 (m, 2H, CH_2_), 3.33 (t, *J* = 1.2 Hz, 2H, 9-CH_2_), 3.43-3.52 (m, 2H, NHCH_2_), 4.66 (t, *J* = 1.2 Hz, 2H, 6-CH_2_), 5.94 (s, 1H, CH), 6.71 (br s, 2H, NH_2_), 7.11-7.29 (m, 5H, 5CH), 7.42 (t, *J* = 5.6 Hz, 1H, NH). ^13^C NMR (75.462 MHz, DMSO-*d_6_*/CCl_4_, 1/3, *v*/*v*) δ_C_: 11.7, 13.1, 26.3, 35.4, 36.3, 40.5, 59.4, 68.7, 100.0, 106.3, 120.9, 124.1, 125.4, 127.7, 128.3, 139.2, 140.3, 143.0, 146.9, 147.0, 147.9, 154.6, 164.6. Anal. calcd for C_26_H_29_N_5_O_2_S: C 65.66; H 6.15; N 14.73; S 6.74%. Found: C 65.49; H 6.24; N 14.57; S 6.86%.

#### 3.1.4. Method for the Synthesis of 5-(3,5-Dimethyl-1*H*-pyrazol-1-yl)-2,2-dimethyl-1,4-dihydro-2*H*-pyrano[4″,3″:4′,5′]pyrido[3′,2′:4,5]thieno[3,2-*d*]pyrimidin-8(9*H*)-one (**4a**)

The mixture of compound 3a (0.8 g, 2.0 mmol) and formamide (5 mL) was refluxed for 4 h. After cooling, the obtained brown crystals were filtered off, washed with water, dried and recrystallized from dimethyl sulfoxide (DMSO). Yield 82%, m.p. > 340 °C. IR ν/cm^−1^: 3340 (NH), 1670 (C=O).^1^H NMR (300 MHz, DMSO-*d_6_*/CCl_4_, 1/3, *v*/*v*) δ_H_:1.38 (s, 6H, C(CH_3_)_2_), 2.25 (s, 3H, CH_3_), 2.43 (s, 3H, CH_3_), 3.54 (t, J = 1.2 Hz, 2H, 1-CH_2_), 4.80 (t, J = 1.2 Hz, 2H, 4-CH_2_), 5.99 (s, 1H, CH), 8.15 (s, 1H, NH), 12.82 (s, 1H, CH). ^13^C NMR (75.462 MHz, DMSO-*d_6_*/CCl_4_, 1/3, *v*/*v*) δ_C_: 12.0, 13.1, 26.4, 36.9, 59.6, 68.7, 107.2, 122.3, 122.9, 124.5, 140.9, 144.9, 146.4, 147.8, 148.4, 151.7, 156.9, 157.5. Anal. calcd for C_19_H_19_N_5_O_2_S: C 59.82; H 5.02; N 18.36; S 8.41%. Found: C 59.98; H 4.95; N 18.51; S 8.57%.

#### 3.1.5. General Method for the Preparation of Compounds **4b**–**f**

A mixture of the appropriate compounds **3c**–**f** (2.0 mmol), triethylorthoformate (2.0 mL) and acetic anhydride (2.0 mL) was refluxed for 2 h. The obtained crystals were filtered off, washed with water, dried and recrystallized from a mixture of chloroform/ethanol 3:1.

*5-(3,5-Dimethyl-1H-pyrazol-1-yl)-2,2-dimethyl-9-phenyl-1,4-dihydro-2H-pyrano[4″,3″:4′,5′]pyrido[3′,2′:4,5]thieno[3,2-d]pyrimidin-8(9H)-one* (**4b**). It was obtained as white crystals; yield 92%, m.p. 238–239 °C. IR ν/cm^−1^: 1671 (C=O). ^1^H NMR (300 MHz, DMSO-*d_6_*/CCl_4_, 1/3, *v*/*v*) δ_H_:1.40 (s, 6H, C(CH_3_)_2_), 2.27 (s, 3H, CH_3_), 2.47 (s, 3H, CH_3_), 3.56 (t, *J* = 1.2 Hz, 2H, 1-CH_2_), 4.84 (t, *J* = 1.2 Hz, 2H, 4-CH_2_), 6.01 (s, 1H, CH), 7.50-7.65 (m, 5H, 5CH), 8.40 (s, 1H, CH). ^13^C NMR (75.462 MHz, DMSO-*d_6_*/CCl_4_, 1/3, *v*/*v*) δ_C_: 12.2, 13.1, 26.4, 37.0, 59.8, 68.6, 107.3, 122.4, 122.8, 124.1, 126.7, 128.6, 128.9, 136.4, 140.9, 144.8, 147.7, 148.1, 148.4, 150.5, 155.6, 157.9. Anal. calcd for C_25_H_23_N_5_O_2_S: C 65.63; H 5.07; N 15.31; S 7.01%. Found: C 65.81; H 5.13; N 15.47; S 6.88%.

*9-(2,4-Dimethoxyphenyl)-5-(3,5-dimethyl-1H-pyrazol-1-yl)-2,2-dimethyl-1,4-dihydro-2H-pyrano[4″,3″:4′,5′]pyrido[3′,2′:4,5]thieno[3,2-d]pyrimidin-8(9H)-one* (**4c**). It was obtained as white crystals; yield 91%, m.p. 285–286 °C. IR ν/cm^−1^: 1668 (C=O). ^1^H NMR (300 MHz, DMSO-*d_6_*/CCl_4_, 1/3, *v/v*) δ_H_:1.40 (s, 6H, C(CH_3_)_2_), 2.27 (s, 3H, CH_3_), 2.45 (s, 3H, CH_3_), 3.57 (t, *J* = 1.1 Hz, 2H, 1-CH_2_), 3.84 (s, 3H, OCH_3_), 3.90 (s, 3H, OCH_3_), 4.84 (t, *J* = 1.1 Hz, 2H, 4-CH_2_), 6.01 (s, 1H, CH), 6.65 (dd, *J* = 8.7, 2.6 Hz, 1H, CH), 6.73 (d, *J* = 2.6 Hz, 1H, CH), 7.29 (dd, *J* = 8.7 Hz, 1H, CH), 8.15 (s, 1H, CH). ^13^C NMR (75.462 MHz, DMSO-*d_6_*/CCl_4_, 1/3, *v*/*v*) δ_C_: 12.1, 13.1, 26.4, 36.9, 55.0, 55.5, 59.8, 68.6, 99.1, 104.8, 107.2, 117.7, 122.3, 122.9, 124.3, 129.1, 140.8, 144.8, 147.9, 148.3, 149.0, 150.5, 155.0, 155.7, 157.9, 161.2. Anal. calcd for C_27_H_27_N_5_O_4_S: C 62.65; H 5.26; N 13.53; S 6.20%. Found: C 62.79; H 5.17; N 13.37; S 6.33%.

*9-(3,4-Dichlorophenyl)-5-(3,5-dimethyl-1H-pyrazol-1-yl)-2,2-dimethyl-1,4-dihydro-2H-pyrano[4″,3″:4′,5′]pyrido[3′,2′:4,5]thieno[3,2-d]pyrimidin-8(9H)-one* (**4d**). It was obtained as white crystals; yield 89%, m.p. 191–192 °C. IR ν/cm^−1^: 1675 (C=O). ^1^H NMR (300 MHz, DMSO-*d_6_*/CCl_4_, 1/3, *v*/*v*) δ_H_:1.40 (s, 6H, C(CH_3_)_2_), 2.26 (s, 3H, CH_3_), 2.47 (s, 3H, CH_3_), 3.56 (t, *J* = 1.1 Hz, 2H, 1-CH_2_), 4.85 (t, *J* = 1.1 Hz, 2H, 4-CH_2_), 6.01 (s, 1H, CH), 7.55 (dd, *J* = 8.6, 2.5 Hz, 1H, CH), 7.75 (d, *J* = 8.6 Hz, 1H, CH), 7.84 (d, *J* = 2.5 Hz, 1H, CH), 8.45 (s, 1H, CH). ^13^C NMR (75.462 MHz, DMSO-*d_6_*/CCl_4_, 1/3, *v*/*v*) δ_C_: 12.2, 13.1, 26.4, 37.0, 59.8, 68.6, 107.3, 107.4, 122.5, 124.0, 126.8, 129.1, 130.6, 132.0, 132.4, 135.7, 141.0, 144.9, 147.4, 148.2, 148.5, 150.5, 155.4, 157.9. Anal. calcd for C_25_H_21_Cl_2_N_5_O_2_S: C 57.04; H 4.02; N 13.30; S 6.09%. Found: C 57.23; H 3.96; N 13.16; S 6.26%.

*5-(3,5-Dimethyl-1H-pyrazol-1-yl)-2,2-dimethyl-9-(2-phenylethyl)-1,4-dihydro-2H-pyrano[4″,3″:4′,5′]pyrido[3′,2′:4,5]thieno[3,2-d]pyrimidin-8(9H)-one* (**4e**). It was obtained as white crystals; yield 84%,m.p. 175–176 °C. IR ν/cm^−1^: 1673 (C=O).^1^H NMR (300 MHz, DMSO-*d_6_*/CCl_4_, 1/3, *v*/*v*) δ_H_:1.38 (s, 6H, C(CH_3_)_2_), 2.25 (s, 3H, CH_3_), 2.44 (s, 3H, CH_3_), 3.11 (t, *J* = 7.2 Hz, 2H, NCH_2_CH_2_), 3.48 (t, *J* = 1.2 Hz, 2H, 1-CH_2_), 4.35 (t, *J* = 7.2 Hz, 2H, NCH_2_CH_2_), 4.81 (t, *J* = 1.2 Hz, 2H, 4-CH_2_), 6.00 (s, 1H, CH), 7.17–7.31 (m, 5H, 5CH), 8.20 (s, 1H, CH). ^13^C NMR (75.462 MHz, DMSO-*d_6_*/CCl_4_, 1/3, *v*/*v*) δ_C_: 12.1, 13.1, 26.3, 34.4, 36.9, 47.6, 59.7, 68.5, 107.1, 122.2, 124.2, 126.2, 128.0, 128.4, 137.0, 140.8, 144.7, 147.9, 148.3, 148.4, 150.9, 156.0, 157.8. Anal. calcd for C_27_H_27_N_5_O_2_S: C 66.78; H 5.60; N 14.42; S 6.60%. Found: C 66.94; H 5.67; N 14.28; S 6.76%.

### 3.2. Biological Evaluation

Compounds were studied for their possible neurotropic activities (anticonvulsant, sedative and antianxiety activity) as well as side effects on 450 white mice of both sexes weighing 18–24 g and 50 male rats of the Wistar line weighing 120–140 g. All groups of animals were maintained at 25 ± 2 °C in the same room on a common food ration. All the biological experiments were carried out in full compliance with the European Convention for the Protection of Vertebrate Animals used for Experimental and other Scientific Purposes. All animal procedures were performed in accordance with the Guidelines for Care and Use of Laboratory Animals of “(ETS No 123, Strasbourg, 03/18/1986): Strasbourg (France). European Treaty Series—No 123, 18 March 1986. 11 P”. University and Experiments were approved by the Animal Ethics Committee of the Scientific Technological Center of Organic and Pharmaceutical Chemistry of the National Academy of Sciences of the Republic of Armenia. P.N5 from 24 March 2016.

#### 3.2.1. Evaluation of the Anticonvulsant Activity of the Synthesized Compounds

The anticonvulsant effect of the new synthesized compounds was investigated by PTZ convulsion tests (Acros organics, New Jersey, USA), MES [22,23,24,25,26]. The PTZ test is an experimental model for inducing myoclonic seizures as well as for predicting the anxiolytic properties of compounds. Outbred mice (weight 18–22 g) were used for the study. The PTZ test was carried out in mice by subcutaneous administration of analeptic at a dose of 90 mg/kg and the effectiveness of the preparations was determined by the prevention of clonic seizures. The anticonvulsant activity of the compounds was also carried out to prevent the tonicextensor phase of the convulsive seizure caused by maximal electroshock (MES). The parameters of the maximal electroshock were 50 mA, the duration was 0.2 s, the oscillation frequency was 50 imp/s, and the evaluation criterion was the warning of the tonicextensor phase of a convulsive seizure. Substances were administered intraperitoneally in doses of 10, 25, 50, 75 and 100 mg/kg in suspension with carboxymethylcellulose (“Viadi—Ingredients”) with Tween-80 (“Ferak Berlin”) 45 min before the injection of the convulsive agent PTZ causing electrical irritation. The control animals were administered an emulsifier. Each dose of compounds for each test was studied in 8 animals. Analogues for comparison were an anticonvulsant drug from the group of succinimide ethosuximide (neuraxpharmArzneimittel GmbH (Germany) [28]. Thecomparison drug, ethosuximide, was administered intraperitoneally in doses from 100 to 300 mg/kg.

#### 3.2.2. Evaluation of the Psychotropic Properties of the Synthesized Compounds

The psychotropic properties of selected compounds (3b–f and 4b–e) were studied by tests: ″open field″, ″elevated plus maze—EPM″ and ″forced swimming″.

*Open field test*. The motor behavior of rats was studied on a modified ″open field″ model [32,33,34]. For this purpose, an installation was used, the bottom of which was dvided into squares with holes (cells). Experiments were performed in the daytime with natural light. Within 5 min of the experiment, the indicators of sedative and activating behavior were determined—the number of horizontal movements, standing on the hind legs (vertical movements), sniffing of the cells. The number of animals on this model was 8 for each compound, control and reference drug. The studied compounds were administered to rats in the most effective dose of 50 mg/kg intraperitoneally as a suspension with methylcarboxycellulose with Tween-80.

*Elevated plus maze—EPM test.* Antianxiety and sedative effects were studied in mice on an ″elevated plus maze″ model [35]. The labyrinth is a cruciform machine raised above the floor, having a pair of open and closed sleeves opposed to each other. Normal animals prefer to spend most of their time in the closed (dark) sleeves of the labyrinth. The anxiolytic effect of the compounds was estimated by the increase in the number of entries into open (light) sleeves and the time spent in them without increasing total motor activity. This model records the time spent in the closed sleeve and the number of attempts to enter the installation center. In the above model, the test compounds and the reference drug were injected intraperitoneally before the experiments. The control animals were administered an emulsifier. Results were processed statistically (*p* ≤ 0.05).

*Forced swimming test.* To assess ″despair and depression″, the model ″compelling swimming″ [39] was used. Experimental animals were forced to swim in a glass container (height 22 cm, diameter 14 cm), filled 1/3 with water. Intact mice swim very actively, but soon they will be forced to immobilize. The latent period of immobilization, the total duration of active swimming and the period of immobilization are fixed for 6 min. The experiments were conducted under natural light.

#### 3.2.3. Evaluation of Incoordination of Movements in the Rotating Rod Test

The adverse neurotoxic (muscle relaxant) effect of compounds was studied in doses of 50 to 100 mg/kg when administered intraperitoneally, as were reference drugs in effective anticonvulsant doses. Myorelaxation was investigated by the test of a ″rotating rod″ in mice [22,29]. To this end, mice were planted on a metal rod with a corrugated rubber coating, which rotated at a speed of 5 revolutions per minute. The number of animals that could not stay on it for 2 min was determined. To determine the ED_50_ and neurotoxic TD_50_, the statistical method of penetration by Litchfield and Wilcoxon was used [30,31]. Maximal tolerated doses (MTD) were also studied. The compounds, as administered by intraperitoneal injection in doses from 500–1800 mg/kg, were investigated.

### 3.3. Docking Studies

#### 3.3.1. Design of Molecular Models

To create three-dimensional molecular models of the studied compounds, the ChemOffice program version 13.0 was used [51]. Minimization and stabilization of the obtained 3D structures were performed by using force fields MM2, which are used to optimize small molecule models [52]. The molecular models of the studied compounds were saved in *.PDB format after optimization. Molecular models of the studied targets were taken from the RCSB [53] with identification numbers SERT transporter [PDBID:5I6X], GABA_A_ receptor [PDBID:4COF] and 5-HT_1A_ receptor [PDBID:3NYA].

#### 3.3.2. Molecular Docking

The AutoDock Vina and AutoDock Tools software packages were used for carrying out molecular docking using the “blind method” [54]. The statistical reliability of the docking results was provided by 10-fold repeatability of 20 initial conformations for each compound with a spatial search volume not exceeding 27,000 Å. The exhaustiveness equally is 200. The choice of the best conformers was carried out on the base of mean square deviation during complexation RMSD ≤ 2Å.

#### 3.3.3. Binding Constant Calculation

To determine the binding constant [55] of the studied compounds with targets, the following equations were used:$$\Delta G_{\exp}=-RT\ln\left(\frac{1}{K}\right)$$
where Δ*G*_exp_—interaction energy, *R*—gas constant, *T*—absolute temperature, *K*—binding constant.

#### 3.3.4. Conformational Analysis and Visualization

To identify the types of binding during complexation of the studied compounds with targets, the Discovery Studio Visualizer [56] was used. The calculated criteria for the radius of interaction were calculated according to the standard: hydrogen bond length 3.4 Å, for Coulomb interactions—9Å and for van der Waals interactions—14 Å.

#### 3.3.5. Statistical Analysis

The cluster analysis of the spatial and energy values of the ligand target complexation was carried out by the k-means method using the ClastVis online instrument [57]. Statistical analysis of the results of the investigation was conducted on the basis of the complexed application of standard statistical methods, including the calculation of standard deviations, average values and standard average errors.

## 4. Conclusions

Pyrazolo substituted new heterocyclic systems—tricyclic *N*-alkyl(aryl)derivatives of pyrano[4,3-*d*]thieno[2,3-*b*]pyridines and tetracyclic pyrano[4″,3″:4′,5′]pyrido[3′,2′:4,5] thieno[3,2-*d*]pyrimidin-8(9*H*)-ones were synthesized by newly developed methods and their neurotropic activity was studied. Compounds were tested for their anticonvulsive action by evaluating the antagonism between the PTZ convulsive action and maximal electrical shock seizures. The evaluation of anticonvulsant activity of all the synthesized compounds revealed that among all tested, compounds **3b**–**f** and **4b**–**e** had a pronounced anticonvulsant action. In the case of tricyclic pyranothienopyridines, the activity depends on the alkylated groups with activity that are ordered as: **3e** ≥ **3b** ≥ **3d** ≥ **3c** ≥ **3f** > **3a**. Similar activity is observed in the case of tetracyclic thienopyrimidines. The selected nine compounds appeared to be more active than ethosuximide according to the test on PTZ. These compounds were also studied for their possible anxiolytic and antidepressant activities by ″open field″, ″elevated plus maze″ and ″forced swimming″ tests. According to the open field behavioral model, all selected compounds, especially compounds **3b**, **3d**, **3e**, **3f** and **4e**, in a statistically significant manner compared to the control, increase the number of sniffing cell examinations, which may be an indication of the antianxiety activity of the compounds. According to the EPM model, all selected compounds increase the time spent by experienced animals in the center, which indicates sedative activity. This is especially is expressed in compounds **3e**, **3f** and **4c**. The results on the EPM model also confirmed the anxiolytic effect of all tested compounds. This effect was especially expressed in compounds **3e**, **3d**, **3f** and **4d**. According to the ″forced swimming″ model, some of the selected compounds (**3e**, **3f** and **4d**) statistically significantly increase the duration of the latent period of the first immobilization and show some antidepressant effect in the same manner as diazepam. The remaining compounds decrease the total time of active swimming. In terms of therapeutic effect, in vivo and in silico results are in accordance.

On the basis of obtained results, ″lead compounds″, which exhibit the best results of interaction with the studied targets according to the initial assessment, were selected. Docking analysis and obtained biophysics properties of complexations show that ″lead compounds″ (**3a**, **3c**, **3d**, **3e**, **4b**, **4c** and **4d**) interact with the GABA_A_ receptor more strongly than comparative drug diazepam. Moreover, **3d**, **4b**, **4c** and **4d** interact in subsite 3 of the ECD, unlike diazepam. Compound **4b** exhibits a directed action without affecting the central binding site of SERT and predominantly hydrophobic interactions with practically all residues are observed, which form an allosteric binding site. Compounds **3c**, **3d**, **3e**, **3f** and **4e** practically bind with the groove of T4L dopamine of 5HT_1A and block it completely.

## Data Availability

The study did not report any data.

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
