# Peer review of "Evaluation of Neurotropic Activity and Molecular Docking Study of New Derivatives of pyrano[4″,3″:4′,5′]pyrido[3′,2′:4,5]thieno[3,2-d]pyrimidines on the Basis of pyrano[3,4-c]pyridines"

_molecules, 2022, doi:10.3390/molecules27113380_

Round 1
Reviewer 1 Report
This manuscript describes the synthesis, biological evaluation and molecular docking studies of a series of pyrano[4'',3'':4',5']pyrido[3',2':4,5]thieno[3,2-d]pyrimidines analogues based on pyrano[3,4-c]pyridines. The finding in this manuscript is interesting, and would be helpful to the relevant research field. However, major revisions are needed to address the following major concerns before acceptance for publication in Molecules.
- The abstract is too long, need to concise.
- The first paragraph, in the introduction section, references are needed in order to give more information.
- In the introduction section, the background was not sufficient summarized. Especially, the relationship between the neurotropic activity and the GABAA receptors and the serotonin transporter.
- This manuscript was poor presented, e.g. Line 79, it’s unclear the meaning of “A(1) adenosine receptor antago-”; From line 135 to 137, it’s unclear the meaning of “The latest was also synthesized in one step from pyridinethione 1 and ethyl chloroacetate in…”; In the table 1, it’s unclear the meaning of “ A 81/ B 78”
- Figures were too blurry to see
- More information about the experimental animals needs to be included, e. g. the numbers per group, types and sex.
Author Response
Reviewer 1
Question Answer
The abstract is too long, need to concise. In our opinion abstract is corresponding by manuscript content.
The first paragraph, in the introduction section, references are needed in order to give more
information. Accepted
In the introduction section, the background was not sufficient summarized. Especially, the
relationship between the neurotropic activity and the GABAA receptors and the serotonin
transporter. In the ″2.3. Molecular Docking″ probably showed the relationship between the neurotropic avtivity and the the GABAA receptors and the serotonin
transporter.
This manuscript was poor presented, e.g. Line 79, it’s unclear the meaning of “A(1) adenosine
receptor antago-”; From line 135 to 137, it’s unclear the meaning of “The latest was also
synthesized in one step from pyridinethione 1 and ethyl chloroacetate in…”; In the table 1, it’s
unclear the meaning of “ A 81/ B 78” “A(1) adenosine receptor antagonists ” it is a type of biological activity and given citation of article (citation â„– 5).
Compound 3a synthesized by two methods (Method A and Method B). By method B target compound (3a) synthesized without separation alkyl-product (2) (these reactions called one-pot).
In the table 1 “ A 81/ B 78” means the yield of compound 3a by method A and method B, which added in the table 1.
Figures were too blurry to see Visualization of the interaction of compounds and diazepam with receptors have been done as it was possible
More information about the experimental animals needs to be included, e. g. the numbers per
group, types and sex. Information about the experimental animals are brought in ″3.2. Biological Evaluation″ section.
Reviewer 2 Report
In this article, new derivatives of pyrano[4'',3'':4',5']pyrido[3',2':4,5]thieno [3,2-d]pyrimidines on the basis of pyrano[3,4-c]pyridines have been synthesized and their neurotropic activity has been screened as well.
Comments and Suggestions to the authors
In the abstract:
- “a new its type” should be grammatically corrected.
- The words “of” and “are” in this sentence “The search of the new effective neurotropic drugs in the series of derivatives of heterocycles containing pharmacophore groups in the organic, bioorganic and medical chemistry are actual problem” should be changed to “for” and “is” respectively.
- In this sentence “… as well as some psychotropic effect.”, “effect” should be changed to “effects”.
- In this sentence “In recent years, there in the level of …”, “in” should be changed to “is”.
In Introduction:
- “In the same time” and “are” in this sentence “In the same time, the number of malignant neoplasms and infectious diseases are increasing year by year.” are better to be changed to “At the same time” and “is”.
- “is” in this sentence “In this regard, the search and study of anticonvulsants possessing the combined psychotropic properties is of unquestionable interest.” should be changed “to are”.
- “For achievement the maximal result” should be changed to “For achievement of the maximal result”.
- “exert a universal effects:” should be grammatically corrected.
- “a considerable interest”, there is no need to use “a”. Because “interest” is uncountable.
The abstract is long. It should be concise.
The quality of Figures 2-5 is low.
The text must be corrected for punctuation.
This sentence in the abstract “Chemical compounds containing two or more pharmacophore groups due to additional interactions with active receptor centers usually enhance biological activity and even lead to a new its type.” should be more explained in the introduction by citing relevant review articles such as https://www.eurekaselect.com/article/110969
Indeed, the hybrid approach is an innovative and powerful synthetic tool for the synthesis of two or more distinct entities in one molecule with novel biological activities.
After this sentence "The derivatives of condensed pyridines are of interest ... ", to more display the significance of condensed pyridine, please see and cite this review article: https://link.springer.com/article/10.1007/s11030-022-10406-8
Author Response
Reviewer 2
Question Answer
“a new its type” should be grammatically corrected. Accepted
2. The words “of” and “are” in this sentence “The search of the new effective neurotropic drugs in
the series of derivatives of heterocycles containing pharmacophore groups in the organic,
bioorganic and medical chemistry are actual problem” should be changed to “for” and “is”
respectively. Accepted
3. In this sentence “… as well as some psychotropic effect.”, “effect” should be changed to “effects”. Accepted
4. In this sentence “In recent years, there in the level of …”, “in” should be changed to “is”. Accepted
In Introduction
1. “In the same time” and “are” in this sentence “In the same time, the number of malignant
neoplasms and infectious diseases are increasing year by year.” are better to be changed to “At
the same time” and “is”. Accepted
2. “is” in this sentence “In this regard, the search and study of anticonvulsants possessing the
combined psychotropic properties is of unquestionable interest.” should be changed “to are”. Accepted
3. “For achievement the maximal result” should be changed to “For achievement of the maximal
result”. Accepted
4. “exert a universal effects:” should be grammatically corrected. Accepted
This sentence in the abstract “Chemical compounds containing two or more pharmacophore groups due
to additional interactions with active receptor centers usually enhance biological activity and even lead to
a new its type.” should be more explained in the introduction by citing relevant review articles such as
https://www.eurekaselect.com/article/110969 Accepted
After this sentence "The derivatives of condensed pyridines are of interest ... ", to more display the
significance of condensed pyridine, please see and cite this review article:
https://link.springer.com/article/10.1007/s11030-022-10406-8 Accepted
Round 2
Reviewer 1 Report
All my concerns have been well addressed, the manuscript can be published as it is.